# Effects of the Italian Law on Patient Safety and Health Professional Responsibilities Five Years after Its Approval by the Italian Parliament

**DOI:** 10.3390/healthcare11131858

**Published:** 2023-06-26

**Authors:** Giuseppe Candido, Fidelia Cascini, Peter Lachman, Micaela La Regina, Chiara Parretti, Valentina Valentini, Riccardo Tartaglia

**Affiliations:** 1Department of Engineering Sciences, Guglielmo Marconi University, 00193 Rome, Italy; g.candido@unimarconi.it (G.C.); c.parretti@unimarconi.it (C.P.); ri.tartaglia@unimarconi.it (R.T.); 2Section of Hygiene and Public Health, Department of Life Sciences and Public Health, Università Cattolica del Sacro Cuore, 00168 Rome, Italy; fidelia.cascini1@unicatt.it; 3Department of Quality Improvement, Royal College of Physicians of Ireland, D02 X266 Dublin, Ireland; peterlachman@rcpi.ie; 4S.C. Clinical Governance and Risk Management, Ligurian Health and Social Care Company 5, Via Fazio 30, 19121 La Spezia, Italy; 5Independent Researchers, 00192 Rome, Italy; v.valentini@alteralex.com

**Keywords:** public health, patient safety, quality of care, health policies

## Abstract

The application of the Italian law No. 24/2017, which focused on patient safety and medical liability, in the Italian National Health Service has been evaluated by a survey conducted five years after the promulgation of the law. The law required the establishment of healthcare risk management and patient safety centers in all Italian regions and the appointment of a Clinical Risk Manager (CRM) in all Italian public and private healthcare facilities. This study demonstrates that five years after the approval of the law, it has not yet been fully implemented. The survey revealed a lack of adequate permanent staff in all the Regional Centers, with two employees on average per Center. Few meetings were held with the Regional Healthcare System decision-makers with less than four meetings per year. This reduces the capacity to carry out functions. In addition, the role of the CRMs is weak in most healthcare facilities. More than 20% of CRMs have other roles in the same organization. Some important tasks have reduced application, e.g., assessment of the inappropriateness risk (reported only by 35.3% of CRM) and use of patient safety indicators for monitoring hospitals (20.6% of CRM). The function of the Regional Centers during the COVID-19 pandemic was limited despite the CRMs being very committed. The CRMs units undertake limited research and have reduced collaboration with citizen associations. Despite most of the CRMs believing that the law has had an important role in improving patient safety, 70% of them identified clinicians’ resistance to change and lack of funding dedicated to implementing the law as the main barriers to the management of risk.

## 1. Introduction

On 8 March 2017, Italy enacted a law on patient safety and medical liability [1]. The first article states: “La sicurezza delle cure è parte costitutiva del diritto alla salute ed è perseguita nell’interesse dell’individuo e della collettività”, meaning that patient safety is part of the right to health of the person, and it is promoted for the public interest through the provision of any healthcare service.

After Denmark in 2004, Italy is the second European country to have a comprehensive law on patient safety, correlating patients’ rights to safe care, protection of health professionals, transparency of processes and outcomes, and fair compensation in the event of harm.

The law introduces important innovations aimed at improving the quality of care. The British Medical Journal (BMJ) published an editorial to mark the law’s promulgation [2]. The key organizational requirement is the establishment of a center for healthcare risk management and patient safety in every Italian region. Other requirements include:The collection of data on risks and adverse events from public and private healthcare and social facilities. The data are to be transmitted to the National Observatory on Best Practices for Patient Safety (Osservatorio Nazionale per le buone pratiche sulla sicurezza in sanità), a body established by the law at the Italian National Agency for Regional Healthcare Services (Age.na.s.).The appointment of a Clinical Risk Manager (CRM) in all Italian public and private healthcare facilities, with defined professional requirements and responsibilities.The introduction of guidelines and safe practices recognized by Italy’s Istituto Superiore di Sanità (National Institute of Health) as a tool for the protection of healthcare professionals in the event of an adverse outcome. Legal action against healthcare professionals is allowed only in cases of malice or gross negligence.The creation of a safe “environment” with the reporting and learning system for clinicians where “the results of a significant event audit cannot be acquired or used as part of legal actions against healthcare professionals”.

The law stipulates that the functions of Regional Centers are essential to collect data on claims and adverse events through reporting systems. In reality, depending on their organizational capacity and staffing, they can also perform other functions to a lesser extent, i.e., training, development of safety practices, promotion of prevention campaigns, and research.

The responsibilities of the CRMs of the peripheral facilities under the law include:assessment of the risk of diagnostic and therapeutic inappropriateness in clinical pathways;analysis of adverse events and claims through audits for significant events and mortality and morbidity meetings;management of reporting and learning systems;staff training in patient safety;implementation of safety practices;support to hospital legal departments;preparation of an annual report [1].

Unfortunately, the law did not provide any financial investment for the implementation of these new roles and functions. This is a limitation to its implementation. Managers have to apply the requirements of the law using their annual budget without any additional resource [1].

Law no.24/2017 also deals substantially with healthcare professional liability and related themes of fault (Article 6), classification of the kinds of liability for the healthcare providers or professionals (Article 7), and possible reparation and compensations (Article 9) [3].

Following the enactment of the law, difficult questions concerning the legal aspects of professional responsibilities remain. These have attracted renewed interest and criticism in the Italian courts and legal literature. Several bills on the topic have been presented in parliament and these have been merged into a single text entitled “Regulations for healthcare and patient safety and for the professional responsibility of healthcare providers”. As these issues remain unresolved, we have decided not to deal with the aspects related to the complex system of the Italian civil and penal code. We also have excluded insurance issues, which are in dispute between lawyers and judges [4]. 

Two years after the law was enacted, a national survey was conducted to evaluate the impact of the law on healthcare workers (HCWs). Four hundred forty-five subjects participated. The results highlighted that the application of the law was significantly and positively related to knowledge and communication of adverse events and sentinel events, checklist adoption, and participation in educational activities on healthcare risk management. The law’s implementation and promotion had been a reliable educational tool to enhance patient safety culture and to involve HCWs in risk management activities. As knowledge of the law, related education, and understanding of its application were still inadequate, it was recommended that educational programs regarding patient safety, risk management, and the contents of the law itself should be vigorously promoted to achieve the clinical governance goals [5]. 

Unfortunately, official national data on claims and adverse events to evaluate the impact of clinical risk management in each Italian region are not available. However, on the basis of official reports relating to three regions (Lombardy, Tuscany, and Emilia Romagna), which were the first to establish clinical risk management centers even before the approval of law 24/2017, the number of requests for compensation has progressively decreased since risk management became a constant practice in hospitals. Concurrently the reports of adverse events have increased, in particular near misses and minor accidents, while the number of sentinel events has remained constant [6,7,8]. 

As 2022 marked the 5th anniversary of the law promulgation, we have conducted a new survey of CRMs and Directors of the Regional Centers for the management of healthcare risk in order to assess the grade of adherence to the responsibilities and actions required by the law and the causes of failures of implementation.

## 2. Materials and Methods

### 2.1. Study Design and Setting

A cross-sectional analysis was conducted to collect data on the implementation of the law using two online questionnaires. The first questionnaire focused on Regional Center managers, and the second focused on CRMs of public hospitals in Italy. Distribution of the questionnaire was through an e-mail invitation to participate sent by the “Italia in Salute” Foundation.

Twenty-one Regional Centers (Trentino-Alto Adige Region has been considered as two different autonomous Provinces, Trento and Bolzano) and one hundred and sixty CRMs of public hospitals were contacted to participate in the study. Data collection was conducted from 7 November to 8 December 2022.

To protect the confidentiality of the information, the survey was anonymous. Responses to the questionnaires were collected using the Google^®^ Forms platform only for respondents who gave their written consent to participate in the study.

### 2.2. Key Study Measures

The main objective of the survey of the Regional Centers for Safe Care was to ascertain:how they have implemented the directives and requirements of the law;whether they have a coordinating role at the regional level;and what role they have had in the management of the COVID-19 pandemic [9].The main objective of the survey of CRMs was the aim to understand:how they are organized;what kind and how many team members they have at their disposal;what kind of activities they perform;what challenges face;and what interaction they have with the regional center and other hospitals [10,11].

In designing the questions, we started from the organizational directions given in the law [1]. Completion of both questionnaires took about 15 min.

The questionnaire for the Regional Centers (see Appendix A) consisted of 6 sections that included organization (9 items), essential activities (11 items), monitoring of health care companies (1 item), communication (7 items), training (2 items), and COVID-19 pandemic management (3 items). It included questions such as year of establishment, full-time staff, estimated percentage of total time spent on activities conducted annually by the Center, actions to be taken to facilitate clinical risk management and increase patient safety, and the promotion, updating, or implementation of recommendations and best practices guidelines for safe care.

Furthermore, an attempt was made to elicit the actions undertaken to encourage the implementation of the reporting and learning system of adverse events in all regional structures, direct participation in the analysis of sentinel events, and the indicators with which the Center monitors healthcare organizations annually. Finally, issues were investigated such as: conducting courses or other initiatives on the subject of “difficult communication” (disclosure)[12] and organizing and promoting campaigns on good practices to sensitize professionals, etc.

The questionnaire for CRMs consisted of 29 items (see Appendix A) and was divided into 6 sections that included: general and organizational characteristics (8 items), activities carried out and tools (10 items), difficulties in implementation of Law N° 24/2017 and areas for improving the quality and safety (4 items), training (2 items), interaction with Regional Centers (2 items), and the role played by individual CRMs in managing the COVID-19 pandemic.

## 3. Results

The response rate for Regional Centers was 52% (11 out of 21) and for Clinical Risk Managers 42% (68 out of 160, of which over 50% of respondents were Hygiene and Public Health or Forensic Medicine specialists. The profile of the Regional Centers’ directors was different as 45.4% (five out of eleven) of the coordinators of the Centers were non-medical professionals who have undergone specific training on health risk (e.g., a master’s degree or advanced course) with at least three years of experience.

### 3.1. Regional Centers Survey Results

#### 3.1.1. How Have Regional Centers Implemented the Directives and Requirements of the Law?

There are different ways Regional Centers spend their time to deliver risk management as indicated in Figure 1 and Figure 2. The ranges are 10–20% of their time focused on education and training, up to 20% on liaison with citizens, 0–50% on data analyses for monitoring adverse events and claims, 0–50% on the dissemination and promotion of best practices, and 0–10% on proactive approaches to improve clinical pathways.

However, Regional Centers seem to neglect the importance of clinical procedures’ appropriateness for risk management as only six Centers reported the analysis of clinical appropriateness of care. Five Centers reported activities with citizen associations demonstrating the lack of cooperation with stakeholders such as patient and medical associations. The role of scientific research in the risk management field was not prioritized as only four Centers reported research work.

The lack of sufficiently trained permanent staff is significant in all the Regional Centers, with, on average, only two employees per Center. Doctors and nurses are represented more than other professionals. There are no more than four meetings per year to discuss trends and issues with the Regional Healthcare System. 

The essential activities provided by the Centers to promote patient safety mainly concern the adoption of incident reporting in the healthcare organizations at a regional level, support for educational activities and the involvement of the CRMs at different healthcare providers’ levels to share best practices, and the annual revision of the patient safety regional action plan based on the data analysis of the sentinel events.

A specific question was asked about the perceived priority categories given to control patient safety risks at the level of the Regional Centers. The perceived priorities in order of importance are:(a)Healthcare Acquired Infections (HAI), especially those related to Anti-Microbial Resistance (AMR);(b)Pharmacovigilance and medication-related adverse events;(c)Patient safety in operating rooms, especially obstetric care;(d)Implementation of guidelines;(e)Education of professionals.

#### 3.1.2. Are Regional Centers Playing a Coordinating Role at the Regional Level?

Monitoring activities and measures adopted by the Regional Centers mainly address sentinel events and medical malpractice claims analyses. Nonetheless, 20% of the interviewed Centers do not use indicators for measuring nor monitoring and evaluating activities performance. International comparative measures are not used.

The activities performed most by the Regional Centers Public include communication campaigns addressing different patient safety hot topics such as prevention of HAI and AMR and violence against healthcare workers. The disclosure of patient safety incidents and the dissemination of results through the publication of scientific papers on incidents recurrence remains insufficient.

The patient safety education activities at the Regional Center level involve more than 80% of the healthcare operators; although, less than 10% of managers and directors appear to be interested in patient safety education.

#### 3.1.3. What Role Did Regional Centers Play in the Management of the COVID-19 Pandemic?

Only 54.5% of the Regional Centers (6 out of 11) indicated that they had played an active role in the management of the COVID-19 pandemic. Directors were involved, but not always with a formal intervention. 

Directors of only three Centers were part of the regional task force. Six Centers performed a support function for frontline staff during the pandemic with respect to the measures to be taken for patient safety.

### 3.2. Clinical Risk Managers Survey Results

#### 3.2.1. How Are CRMs Organized and What Human Resources Do They Have?

Sixty-eight of the completed questionnaires were included in the analysis (see Table 1). Just over half of the respondents are female (51.5%), 50% are between 56 and 66 years old, 44.1% work for a local health organization, and 66.8%, have a senior medical position. More than 20% of the CRMs have another role in the organization where they work, such as Hospital Medical Officer, Clinician, etc.

Respondents were from 12 Italian regions and the Autonomous Province of Bolzano. The most representative regions were Lazio (26.5%), Veneto (14.7%), and Tuscany and Emilia-Romagna (10.3%).

Finally, 54.4% of the respondents (37 out of 68) have a post-graduate degree in Legal Medicine or Hygiene and Public Health. The remainder have different degrees or medical specializations. Five CRMs (7.4%) have not undertaken any specific training in clinical risk management (Table 1).

#### 3.2.2. What Kind of Activities Do CRMs Perform?

The activities performed by the CRMs are mainly focused on measurement, as indicated in Table 2. International indicators are used in only 20% of the cases. Incident reports are used as indicators by less than 3% of CRMs. The evaluation of health services’ appropriateness is considered by less than 35% of the CRMs, which is similar to that of the Regional Centers.

Over the past three years, training courses for staff have been held to varying degrees by CRMs. Some centers have held up to 50 courses per year, though the average is 11 per year. Events for the public, e.g., seminars or meetings, were also held. However, 60% of CRMs have held less than nine courses. Few papers have been published in journals indicating a lack of research on patient safety and risk management.

#### 3.2.3. What Challenges Do CRMs Face?

The perceived barriers in performing activities consistent with the CRM role are concentrated into three areas:(a)75% of respondents demonstrated resistance to a cultural change embracing patient safety.(b)72% of respondents reported a shortage of human resources dedicated to the role within the healthcare organization.(c)50% of respondents indicated a lack of awareness of the benefits gained by risk management at the different levels of the organization) (Figure 3).

In this context, the approach to Law n. 24/2017 is controversial. Some of the CRMs agree that the law has been beneficial in that it has defined the responsibilities of healthcare professionals and organizations to improve patient safety and to prevent medical malpractice claims. Other respondents consider the law to be not as complete as it should be, as it does not include penalties, standardized criteria for the evaluation of quality and efficacy of the safety interventions, nor the statutes for its application in the field of medical insurance.

Other criticisms include the absence of clinical guidelines development at a national level to allow the appropriate application of the law and the unclear role of the CRM with no direction on how to perform the responsibilities specifically related to the role.

#### 3.2.4. What Interaction Do CRMs Have with the Regional Center and Other Hospitals?

Finally, 72% of the CRMs do not have regular interaction with the Regional Centers for Patient Safety to update strategies and approaches at a regional level with the aim of standardization of risk management procedures and spreading of best practices. During the pandemic, 85.3% of the CRMs were fully involved in accordance with their skills. Furthermore, 68% of the CRMs were part of the local task force and 82.40% performed a support function for frontline staff with respect to the measures to be taken for the safety of the patients through the dissemination of recommendations [13].

## 4. Discussion

This paper describes the degree of implementation of Law 24/2017 at the regional level and at the organizational level five years after its enactment through a survey of the directors of Regional Centers for clinical risk management and the CRMs of public health organizations belonging to the Italian Network for Safety in Healthcare [10,14].

### 4.1. Distribution of Respondents

The responses to a greater extent come from the north–central regions of Italy, albeit with some exceptions. This finding is similar to that in the survey conducted by FIASO (Federazione Italiana Aziende Sanitarie e Ospedaliere) [10]. It could be attributed to the major economic crisis in southern regions and/or a lower degree of their integration into the national network.

### 4.2. Organization and Staffing

The survey clearly shows a level of organization and efficiency, as well as a degree of adherence to the legislative statutes. However, there is variation between Centers and Regions. This can be the result of the absence of financial support by the Italian State for the implementation of the law’s requirements. This would explain the low staffing levels and multi-tasking performed by CRM within their organizations. Physicians comprise less than 30% of the staffing of Regional Centers. In contrast, 69.4% of CRMs are physicians, of whom 54.4% are specialists in hygiene and public health and forensics [14]. This influences the type of activities conducted by Regional Centers and CRMs.

A greater emphasis has been placed on claims and sentinel events, both in terms of the percentage of time dedicated and the frequency of dealing with CRMs. Less than 20% of Regional Centers publish data on litigation and sentinel events for the public, demonstrating that transparency as required by Law 24/2017 needs to be implemented.

### 4.3. Activities

The Regional Centers focus mainly on training, management of information systems, and sentinel event analysis, usually carried out by clinical consultants. They have limited links to the activities of the CRMs in the peripheral facilities and few are involved in the analysis on the appropriateness of clinical pathways. This could presumably be due to their lack of clinical training which creates a “distance*”* in terms of skillset. Moreover, the location of the Regional Centers within the bureaucratic–administrative apparatus of the Regional Government also influences the content of the work, with little predisposition for research and support of clinical activities, even during the pandemic.

Only 54% of the Regional Centers supported healthcare workers practically during the pandemic in contrast to support provided by 82% of peripheral CRMs. Regional Centers provide suboptimal public communication as only 60% of respondents stated that they have a public web page or carried out communications to citizens on quality and safety.

At the peripheral level, it is striking that most centers only measure sentinel events and litigations (ministerial flows) and very few have active patient safety monitoring which is not required by the Ministry of Health. Only 30% of respondents lead a dedicated risk management unit, thereby demonstrating that clinical risk management is still not recognized as having a strong and strategic value. The frequency of appropriateness analyses by CRMs is also low.

### 4.4. Barriers to Implementation of the Law

75% of CRMs report a lack of human resources dedicated to risk management and 72% reported resistance by clinicians and management to changes promoting safety-oriented behaviors. The training provided by Regional Centers is rarely aimed at top management which could explain the low degree of support for clinical risk management within the healthcare organization and the lack of provision of human and economic resources. The responses also indicate that healthcare professionals do not have the knowledge and awareness of the real impact that the safety law could have. This latter assessment in itself would merit a subsequent investigation into the actual ability of regulations to change behaviors in a healthcare setting. 

### 4.5. Value of the Law

As CRMs work in very different operational and organizational conditions, they have varying opinions about the usefulness of the law. Most agree that it has been useful to provide clarity on accountability and how to improve patient safety. Few believe that the law protects practitioners. In addition, the law does not provide standard criteria for assessing the quality and safety of care.

Limitations of the law as perceived by CRMs include resistance to change and the lack of dedicated human and financial resources. Directors of Regional Centers and facilities CRMs agree that there is a need for standardization of procedures and best practices, benchmarking about clinical risk management activities among different organizations, and analysis of sentinel events.

### 4.6. Solutions and Recommendations

There are lessons other countries can learn from the Italian experience. A legal framework for patient safety is attractive as it provides a focus for patient safety initiatives and, at a policy level, this can influence how safety is achieved.

Based on the result of the survey, in particular from Regional Centers, and recommendations of international institutions [15], a patient safety law will require the following support and actions to be effective:Finance: Financial investment to facilitate implementation of the law, focusing on infrastructure, data management for sharing learning, and human resources to staff regional Centers and organization staffing.Cost effectiveness: Inclusion of the cost-effectiveness of patient safety as postulated by the OECD to validate financial support for legal requirements [16].Education: Training on patient safety at all levels, starting in the medical and nursing colleges to the frontline staff and management at all levels. This should include human factors and ergonomics, resilience and reliability theory, incident investigation, and learning.Culture: A focus on developing a learning patient safety culture with incentives for good performance and penalties for negligent behaviors.Learning system: Learning from daily good practice as well as from adverse events [17].Integration of learning: An independent agency that will take reports about patient safety events, analyze data and look for patterns of risks of harm, investigate, and make recommendations for change [18].

## 5. Limitations

A limitation of our research concerns the response rate of the Regional Centers Directors and CRMs. However, this confirms the prevailing difficulties due to staff shortages and a lack of awareness of their role by the top management. We did not investigate the application of prevention activities established by the law through “output” indicators. We intend to conduct this evaluation in the near future, following from that which we have done in the past and which determined a 5.2% rate of adverse events in five large Italian hospitals [19].

## 6. Conclusions

This survey highlights significant challenges to the implementation of a law on patient safety in Italy with the subsequent impact on the activities aimed to guarantee safety of care in the Italian healthcare system. The results show that, although a law may establish important principles, it is necessary to create a strong organizational system to implement the law and to allocate appropriate resources to fund the implementation. Investment for implementation was not included in the law and this should be rectified to assist the full implementation of the law. Further specification of the skillset for regional directors and peripheral CRMs is required, including a focused training and certification process. Finally, the promotion of research activities in patient safety is essential to make healthcare safer.

## Figures and Tables

**Figure 1 healthcare-11-01858-f001:**
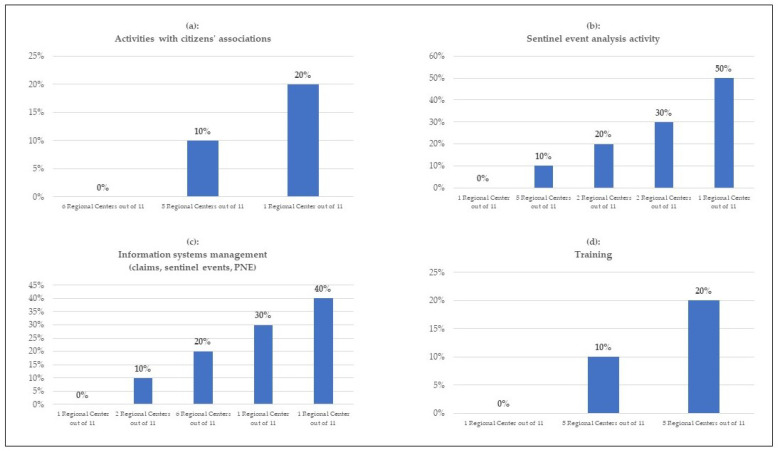
Estimate as a percentage of the total time dedicated to the activities carried out annually by the Regional Centers.

**Figure 2 healthcare-11-01858-f002:**
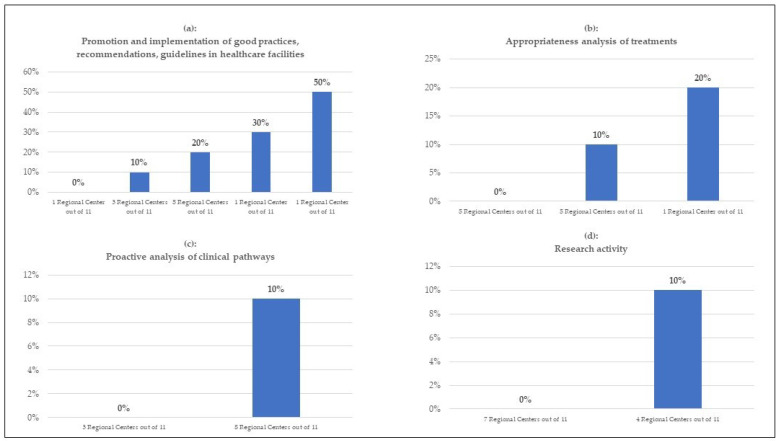
Estimate as a percentage of the total time dedicated to the activities carried out annually by the Regional Centers.

**Figure 3 healthcare-11-01858-f003:**
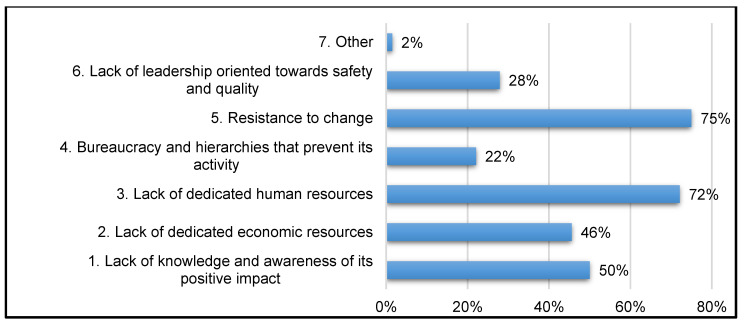
Barriers encountered in carrying out the role.

**Table 1 healthcare-11-01858-t001:** Descriptive analysis of CRM respondents.

Sample Characteristics		N	%
Gender	Male	33	48.5
	Female	35	51.5
Age range	35–55	7	10.3
	46–55	20	29.4
	56–66	34	50
	>66	7	10.3
Workplace region	Veneto	10	14.7
	Autonomous Province of Bolzano	1	1.5
	Abruzzo	3	4.4
	Campania	3	4.4
	Emilia-Romagna	7	10.3
	Friuli Venezia Giulia	4	5.9
	Lazio	18	26.5
	Liguria	5	7.4
	Lombardy	2	2.9
	Marche	2	2.9
	Piedmont	4	5.9
	Tuscany	7	10.3
	Umbria	2	2.9
Work placement	Hospital and University Hospital	17	25
	Local Health Company	30	44.1
	IRCCS *	6	8.8
	Accredited Private Facility	5	7.4
	Other	10	14.7
Professional role	Head of a complex unit	22	32.4
	Head of a simple unit	22	32.4
	Medical Director	19	27.9
	Health Professions Director	3	4.4
	Non-Healthcare Manager	2	2.9
Facility CRM qualification	Specialist in hygiene or forensic medicine	37	54.4
	Physician with another specialty but with training in clinical risk management and at least three years of experience	17	25
	Employee with another degree but who has undergone specific training on health risk management (master’s, advanced course) and has at least three years of experience	9	13.2
	Other manager without specific training in health risk management	5	7.4

* IRCCS (Istituti di Ricovero e Cura a Carattere Scientifico): Institutes for Hospitalization and Treatment of a Scientific Nature.

**Table 2 healthcare-11-01858-t002:** Activities and Tools of CRMs.

Activities and Tools		N	%
Which of the following indicators do you use to monitor the safety of care in your healthcare enterprise?	OCse indicators of patient safety	14	20.6
	PNE * indicators	33	48.5
	Claims	54	79.4
	Sentinel events	62	91.2
	Adverse events	5	7.4
	Incident reporting	2	2.9
	Near misses	4	5.9
	Other	1	1.5
Have any clinical audits been conducted in the past 36 months on the results of PNE *?	Yes	36	52.9
	No	32	47.1
Do you perform analyses to assess the risk of inappropriateness?	Yes	24	35.3
	No	44	64.7
Does your facility have a dedicated page on the facility website to post the annual report and useful information for citizens?	Yes	61	89.7
	No	7	10.3
Have you ever been asked to hand over minutes or other documents related to audits to the judiciary after Law 24/2017 was passed?	Yes	17	25
	No	51	75

* PNE (Programma Nazionale Esiti): National Outcome program calculates the mortality for some medical, surgical, and other invasive procedures in all Italian hospitals.

## Data Availability

Data are available on reasonable request. Requests for data sharing from appropriate researchers and entities will be considered on a case-by-case basis. Interested parties should contact the corresponding author.

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
