# Peer review of "Effects of the Italian Law on Patient Safety and Health Professional Responsibilities Five Years after Its Approval by the Italian Parliament"

_healthcare, 2023, doi:10.3390/healthcare11131858_

Round 1

Reviewer 1 Report

In my opinion the study could be interesting altough, as the Authors stated, the responders get the study itself of limitated significance-

In my opinion the table 1 and 2 must be clarified, evntually using other graphic methods: in fact in the actual form they are comprehesible wih difficulty.

The section solutions and recommendations, according to me, speaks about personal opinions/experiences of the Authors and it is not in right connection with the section results of the study.

In my opinion the Authors must better relate the solutions/recommendations section with the issues they highlight in the previous sections of the paper

Author Response

see the attachment, please

Reviewer 2 Report

1. Page 2/13, line 34 Regarding the statement "Unfortunately, a limitation of the law is that it did not provide any financial investment for the implementation of these new roles and functions. Managers had to apply the requirements of the law using their annual budget without additional resources."

Have your questionnaires considered quantifying how much of the annual budget was allocated for compliance with Law 24? If not, can the reason behind this be explained?

2. Page 9/13, line 8. Reference is made to "The perceived priority categories to control the risks to patients are intended, at the level of the Regional Centres, as follows (in order of perceived priority):

a) Healthcare Acquired Infections (HAI), especially those related to Anti-Microbial Resistance (AMR);

b) Pharmacovigilance and medication-related adverse events;

c) Patient safety in Operating Rooms (especially obstetric care);

d) Implementation of guidance;

e) Education of professionals. 

Was this data extracted from the questionnaires? If yes, can it be presented in the form of a table?"

3. Can a facsimile of the questionnaire sent to the facility managers be included, possibly as an image? From the text, it is not easy to understand what specific questions were asked to the operators.

4. Line 1, page 5. The following text: "However, Regional Centres seem to neglect the impact of clinical procedure appropriateness for risk management, the importance of arranging cooperation with the stakeholders such as patient and medical associations, and the role of the scientific research in the risk management field. The lack of sufficiently trained permanent staff is significant in all the Regional Centres, with two employees per Centre on average. Doctors and nurses are more represented than other professionals. There are no more than four meetings per year to discuss trends and issues with the Regional Healthcare System Directorate General. 

Was it extracted from the questionnaire or is it information from the literature? In this case, provide appropriate clarification and, if applicable, include the appropriate citation.

5. Could you clarify why this type of outcome of Italian law is important at the international level? 

Please provide a more detailed explanation of how these data can be received by the readers of the magazine, perhaps by providing examples comparing the data with other countries or the situation within the European Union.

It is recommended to have it reviewed by a native English speaker.

Author Response

see the attachment, please

Reviewer 3 Report

Pag 1:  In introduction   it is advisable to report the exact expression of the law textually between quotation marks

pag. 2 the sentence "Legal action against healthcare professionals is allowed only in cases of malice or gross negligence" is inadequate. The authors are asked to better specify the conditions governing the legal action against health professionals.

pag. 10 4.5 The sentence "However, the law does not provide penalties for those who do not apply best practices" is inadequate. 

Author Response

see the attachment, please

Round 2

Reviewer 1 Report

I think a lot of the issues I note in the previuos version are still remaining even in the revised versione of the manuscript. The data in the table are now clarified

Author Response

dear editor,

we completely revised the manuscript, according to reviewer's suggestions, even if a bit "vague". 

We made an extensive language revision.

We confirm that the answers received refer to centers who are actually active. We talked about centers existing only the paper as a presumptive explanation of 40% response rate.

Thank you for your attention to our work

best regards

Micaela La Regina
